# Simple deductive reasoning tests and numerical data sets for exposing limitation of today's deep neural networks

## Abstract

Learning for Deductive Reasoning is an open problem in the machine learning world today. Deductive reasoning involves storing facts in memory and generation of newer facts over time. The concept of memory, processor and code in deduction systems is fundamentally different from the purpose and formulation of weights in a deep neural network. A majority of the machine learning models are inductive reasoning models including state of the art deep neural networks which are effectively tensor interpolation based models. A step towards realization of memory is through recurrent neural networks and its variants, however the formal representation is not sufficient enough to capture a complex mapping function between input and output patterns. Deep neural networks are positioned to do away with feature engineering which is essentially deductive reasoning methodology. There are existing works in deductive reasoning in neural networks that require learning of syntax, unification and deduction and operate on text data as sequence of tokens. However the performance of deductive reasoning networks is far from perfection which may be either due to syntax or deduction aspects. In this context, we have proposed a suite of completely numeric data sets which do not require parsing as with text data. The 10 data sets are for - (a) selection (3 data sets) - minimum, maximum and top 2nd element in an array of numbers; (b) matching (3 data sets) - duplicate detection, counting and histogram learning; (c) divisibility tests (2 data sets) - divisibility of two numbers and divisibility by 3; (d) representation (2 data sets) - binary representation and parity. Though extremely simple in terms of feature engineering, in all of these tests, simple deep neural networks, random forest and recurrent neural networks have failed with very low accuracies. We propose these as numerical test-bed for testing learning models for deductive reasoning.

## 1 Introduction

Deductive reasoning is a branch of artificial intelligence where inferences are represented as assertions or facts over data (Khemani, 2013). Starting with a set of given facts, the system combines facts based on rules to generate newer facts and update the knowledge store. On the other hand machine learning algorithms employ induction based approaches which are predominantly pattern mapping methods (McClelland et al., 1986). Fundamentally, in a pipeline of operations, the vectors are arithmetically combined, logically filtered, scaled up or scaled down and mapped to the target vector of interest. A tensor is a more generalization of the vector representation mathematically. However typically even a tensor is internally represented as an array of contiguous storage locations with a data structure indicating dimensions. These tensors undergo a pipeline of transformations minimizing an error function there by mapping a tensor on one side of the pipeline to the tensor on the other side. Deep neural networks have demonstrated their performance almost at the level of human or even better in computer vision and other domains (Bengio et al., 2017).

Although it is promising to see the success of deep neural networks (Dargan et al., 2019) (DNN) there seems to be a popular belief and false opinion that they are suitable for all types of problems. It is important to note here that problem statements solved by DNNs are of mainly of interpolation in nature where tensors are combined along the pipeline to produce the output tensor. The vanilla

DNNs are not directly suitable for *deductive reasoning* type of problem statements. A knowledge base in a deductive reasoning methodology is a storage for facts which are modified over time. For instance, counting number of ones in a binary representation of the input, the current count is a fact and as the algorithm iterates over input, the count value gets updated. Most of the iterations over input, can be represented in a declarative style as first order logic statements such as prolog (Bratko, 2001). The weight space representation of a deep neural network is not a convenient formulation to capture the facts, unification and inference mechanism as required by a deductive reasoning methodology. However, earlier version of machine learning formulations required *feature engineering* which itself accommodates for deductive reasoning in the form of outputs of human crafted programs which are added as features.

There are on-going research efforts in this direction on modification of neural network architectures to enable them for performing deduction reasoning. A small step in the direction of storage of past information in data is a recurrent neural network formulation and its several variants (Mikolov et al., 2011). Most of the existing works employ a recurrent neural network based formulation and its variations due to the fundamental need of the notion of memory in a deductive reasoning setting. We have tabulated the observations in the form of a Table 1.

| Citation | Data | Processing |
|---|---|---|
| (Nangia & Bowman, 2018) | Input is a string of list operations and elements such as minimum and maximum. Output is the result of list operations. They have released a ListOps data set. | TreeLSTM (Tai et al., 2015b) |
| (Saxton et al., 2019) | Input is a stream of sequences of tokens. A specific sequence is defined as a question. The output is again a sequence of tokens corresponding to answer. | They have used an RNN based formulation for a question-answer system. |
| (Wu et al., 2020; Irving et al., 2016; Gauthier et al., 2020; Bansal et al., 2019; Polu & Sutskever, 2020; Evans et al., 2018; Lample & Charton, 2019) | Input is a string of tokens corresponding to a truth statements of a theorem. Output is a string of tokens corresponding to proof for the theorem or identification of top few premises. | They have used a variation of RNN formulation. |
| (Yang & Deng, 2019) | Input is a string of knowledgebase and theorem. Output is a string of proof statements. They also release CoqGym data set of 71K proofs. | A TreeLSTM (Tai et al., 2015b) formulation is used. |
| (Huang et al., 2018) | Input is a string representation of a theorem. Output is estimation of number of steps required to finish and prediction of the next expression. | RNN based formulation. |
| (Piotrowski et al., 2019) | Input is a string of tokens corresponding to polynomial coefficients in symbolic form before normalization. Output is a normalized equivalent expression. | They have used RNN based formulation. |
| (Paliwal et al., 2020) | Input is a string of tokens corresponding to theorem. Output is a string of tokens corresponding to the first step of the proof. | The authors have used Graph Neural Networks (Zhou et al., 2018). |

| (Hahn et al., 2020; Rabe et al., 2020) | Input is a string of tokens corresponding to a representation of arithmetic and logical expression. Output is an equivalent expression in another representation. | Sequence to sequence mapping using RNN based formulation. In (Rabe et al., 2020) they introduce Skip-tree architecture. |
|---|---|---|
| (Xu et al., 2019) | Input: $x_i, y_i$ as numeric, Process: Theoretical study of what neural networks can and cannot do, Output: A theoretical study, Contribution: They suggest max(), subset sum() problems. | |
| (Wang et al., 2019; Xu et al., 2018; Amizadeh et al., 2018; Selsam et al., 2018; Rocktäschel & Riedel, 2016) | Input is a string of tokens corresponding to logical expression. Output is boolean for satisfiability or a string of tokens corresponding to value assignments of variables. | Some of the architectures proposed are (Xu et al., 2018) CSP-CNN, (Amizadeh et al., 2018) DG-DAGRNN, (Selsam et al., 2018) NeuroSAT and RNN based formulations. |
| (Vinyals et al., 2015) | Input is a set of tokens in tuple format (position,token). Output is a sequence of tokens. | RNN based formulation is used to carry out sorting and copy operations. The idea is to represent a sequence as a set. |
| (Devlin et al., 2018; Sukhbaatar et al., 2015; Tai et al., 2015a; Weston et al., 2014) | Input is a sequence of tokens corresponding to a 'question'. Output is another sequence of tokens conrresponding to 'answer'. | Variants of RNN based formulation such as (Devlin et al., 2018) BERT, (Gers et al., 1999) TreeLSTM |
| (Grefenstette et al., 2015) | Input and output are vector sequences. | An RNN based formulation with notion of memory in the form of a stack or queue of hidden vectors. The output is generated as weighted combination of these memory vectors (Neural stack and queue). |
| (Graves et al., 2014) | Input and output are vector sequences. | An RNN to include a controller network and memory network for function selection and operand selection towards building a Neural Turing Machine. |

Table 1: Literature review of the state of the art methods for deductive reasoning in deep learning.

The symbolic reasoning neural networks (Table 1) obtain the input is a variable length string of tokens and the output is also a variable length string of tokens. Some tokens have special meaning corresponding to syntactic and semantic interpretations such as opening and closing brackets, function call and data. While this problem formulation is more generic in nature, the onus of parsing the input token list rests with the DNN in addition to interpretation and output generation.

In a deductive reasoning setting, the concept of memory becomes critical. Some of the deductive reasoning tasks can be easily accomplished by a simple computer program to result in numeric features or truth value statements. However the effect of a free form neural network do the same and without tweaking is far from reach. Basic notion of memory in an RNN formulation is the hidden vector formulation. In order to perform symbolic unification, logical operations and learn a generic function from data, the hidden vector concept is reformulated for a stack or queue of vectors (Grefenstette et al., 2015). The Neural Turing Machine (Graves et al., 2014) includes a controller network and memory network to mimic a Turing machine and attempts to learn generic function that maps input to output from data.

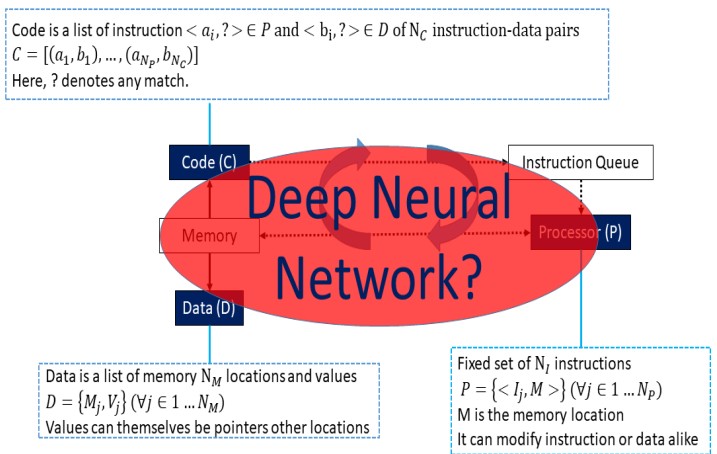

Figure 1: This picture depicts the three critical components of a general purpose mapping function - processor, memory and code. A deep neural network attempts to model all three components in its weight space representation.

In order to learn a generic mapping function from one pattern to another pattern, it is important to see a basic computational model. In (Figure 1) we have indicated a conceptual processor, code and data. A symbolic reasoning network tries to model the three entities simultaneously in the form of weights of the network. A processor has a fixed set of instructions that operate on memory, $P = \{(I_j, M)\}(\forall j \in 1 \ldots N_P)$ where $N_P$ denotes number of instructions, $I_j$ denotes $j^{th}$ instruction, $M$ denotes a memory location. The data is indicated by, $D = \{M_k, V_k\}(\forall k \in 1 \ldots N_M)$ where $N_M$ is the number of locations, $M_k$ is the $k^{th}$ memory location and $V_k$ is the value present at the location. A program or code is indicated by $C = \{(a_i, b_i)\}(\forall i \in 1 \ldots N_C)$ where $a_i$ is the instruction and $b_i$ is the memory location. We do here some back of the envelope calculations on how many essential vectors need to be captured for instructions, memory value pairs and code. If we assume one hot encoding representation of all instructions, this would require an estimate of $N_P \times N_M$ pairs of instructions. For each of the memory locations, the values are typically floating point numbers or integers with large ranges, the possible numbers of memory-value pairs is an astronomical number $N_M \times |V_k|$ where $|V_k|$ denotes value range. Each term in a program would correspond to one-hot-encoding of $N_P \times N_M$ instruction and memory pairs. The number of programs of length $N_C$ that can be generated would be $N_P \times N_M{}^{N_C}$. Even for a simple processor having 10 instructions, 10 memory locations and a program of 10 steps, would require exploration of a space of $100^{10}$ vectors corresponding to each program. It is not clear today, what is the best way to represent and model the process as it is evident from low accuracies and high errors reported in state of the art symbolic reasoning frameworks.

As symbolic processing requires a neural network to *learn text parsing* and also *learn inference* from data. There are two moving parts in order to understand low accuracies observed in deductive tasks. In order to specifically identify issue, we have provided numeric data sets that does not require text parsing. It is also important at this stage of the state of the art, to clearly call out limitations of the existing deep neural networks and machine learning formulations on what is and what is not possible in a more specific way. As machine learning systems operate over data sets, the first step in demonstrating what is not possible is via simple data sets for pattern mapping. We have created some very simple data sets where a single engineered feature is sufficient enough to capture the pattern, however deep networks fail to capture the deduction patterns. The data sets are for - (a) selection (3 data sets) - minimum, maximum and top 2nd element in an array of numbers; (b) matching (3 data sets) - duplicate detection, counting and histogram learning; (c) divisibility tests (2 data sets) - divisibility of two numbers and divisibility by 3; (d) representation (2 data sets) - binary

| S.No | Data set (N=100000, K=50) | Algorithm invocation |
|---|---|---|
| 1 | MIN | SEL("min", N, K) |
| 2 | MAX | SEL("max", N, K) |
| 3 | TOP2 | SEL("top 2", N, K) |
| 4 | BINARY | REP("parity",N,K) |
| 5 | SORT | SORT(N,K) |
| 6 | COUNT | MAT("count",N,K) |
| 7 | DIV | DIV("divisible",N,K) |
| 8 | DIV3 | DIV("divisible_by_3",N,K) |
| 9 | PARITY | REP("parity",N,K) |
| 10 | DUP | MAT("duplicate",N,K) |

Table 2: Data set name and algorithm that generated the data set are shown here.

representation and parity. We demonstrate that in all of these data sets, the deep neural networks fail with very low accuracies.

The efforts in the state of the art for symbolic neural networks mainly include on parsing of the textual input, unification and deduction. These works try to address based on tweaking the architecture of the network for bringing in memory. The RNN based formulation addresses processing part. However, we observe and propose for research there is additional component which is the type of neuron used. In current networks, a single neuron performs only simple arithmetic operations and a comparison against zero as in ReLU. We conjecture here there is scope for innovation in increasing the computational capability of a neuron and ad-hoc connections as proposed in Webster (2012).

## 2 METHODS

In order to present how deductive reasoning is fundamentally different from interpolation based reasoning, we have created ten data sets which are described in this section. Data sets are generated by invoking the algorithm as shown in the (Table 2).

### 2.1 DATA SETS FOR SELECTION PROBLEMS

Identification of maximum or minimum element in an array of numbers, requires facts to be recorded in a storage, infer on top of them in the current iteration and update the storage with newer facts. The data set consists of $D = (x_i, y_i)$ tuples $i \in [1..N]$ where $N = 100000$. Each $x_i \in R^K$ is a $K$ dimensional vector where $K = 50$ in our case and $y_i$ contains the value of an element of interest in the array, i.e. $y_i = max(x_{i,1}, \ldots, x_{i,K})$.

The data set is generated as shown in the schematic (Algorithm 1).

---

**Algorithm 1** Selection Data Set Generation Algorithm - SEL(T,N,K)

---

**Require:** $T, N, K$
  $(\forall i \in [1..N]), (\forall j \in [1..K])x_{i,j} = RAND()$
  /* $RAND()$ function generates a random number */
  /* $T$: Type of selection, $N$: Number of elements in data set, $K$: Dimensionality of each point */
  **if** $T$ is "max" **then**
    $(\forall i \in [1..N]) : y_i = max(x_i)$
  **else if** $T$ is "min" **then**
    $(\forall i \in [1..N]) : y_i = min(x_i)$
  **else if** $T$ is "top 2" **then**
    $(\forall i \in [1..N]) : y_i = max(set(x_i) - \{max(x_i)\})$
  **end if**
  **return** $(x, y)$

---

## 2.2 Data Sets for Matching Problems

The data set is almost same as for the selection problem, however here, duplicate detection and histogram learning are carried out. This requires the numbers to be remembered and compared against other elements. The Duplicate data set contains of $D = (x_i, y_i)$ tuples $i \in [1..N]$ where $N = 100000$. Each $x_i \in R^K$ is a $K$ dimensional vector where $K = 50$ in our case and $y_i$ contains either 1 or 0. 1 if a element $j \in x_i$ is present more than once.

The Histogram data set contains of of $D = (x_i, y_i)$ tuples $i \in [1..N]$ where $N = 100000$. Each $x_i \in R^K$ is a $K$ dimensional vector where $K = 50$ in our case and $y_i$ contains the count of elements which lies between 1 and 10. The data set is generated as shown in the schematic (Algorithm 2).

---

**Algorithm 2** Matching Data Set Generation Algorithm - MAT(T,N,K)

---

**Require:** $T, N, K$

  $(\forall i \in [1..N]), (\forall j \in [1..K]) x_{i,j} = RAND()$

  /* $RAND()$ function generates a random number*/

  /* $T$:Type of Matching, $N$: Number of elements in data set, $K$: Dimensionality of each point */

  **if** $T$ is "duplicate" **then**

    $(\forall i \in [1..N]), y_i = 0, (\forall j, k \in [1..K], j \neq k \ \& \ \& \ x_i[j] == x_i[k]) : y_i = 1$

    **for** $(\forall i \in [1..N])$ **do**

      /* check if a element is present more than one time */

      $y_i = 0$

      **for** $\forall j \in [1..K]$ **do**

        **for** $(\forall k \in [1..K])$ **do**

          $(j \neq k \ \& \ \& \ x_{i,j} == x_{i,k})$

          $\rightarrow y_i = 1$

        **end for**

      **end for**

    **end for**

  **end if**

  **if** $T$ is "count" **then**

    **for** $(\forall i \in [1..N])$ **do**

      //count number of vector elements ¡= 10

      **for** $(j \in [1..K])$ **do**

        $x_{i,j} <= 10 \rightarrow y_i = y_i + 1$

      **end for**

    **end for**

  **end if**

  **return** $(x, y)$

---

## 2.3 Data Sets for Divisibility Problems

For divisibility tests the facts are to be recorded in storage and new element is processed based on this fact and after processing, update the storage with the new fact. For example if we want to perform divisibility test if 1342 is divisible by 11. At first 13 will be evaluated against 11 it will give a remainder 2 which is new fact that we need to store and move on to next element and that's 4. Combining this element with our fact we should get 24 and that will be evaluated against 11 and will give remainder or new fact as 2. And this keeps on going until no element is left.

The Divisibility data set contains of $D = (x_i, y_i)$ tuples $i \in [1..N]$ where $N = 100000$. Each $x_i$ contains a tuple of two elements $x_i = a, b$ and $y_i$ contains either 0 and 1. 1, if $(a \mod b == 0)$ and 0, if $(a \mod b \neq 0)$.

The data set is generated as shown in the schematic (Algorithm 3).

## 2.4 Data Sets for Representation Problems

This contains two data sets $i)$ parity data set and $ii)$ binary representation data set. The parity data set contains of $D = (x_i, y_i)$ tuples where $x_i \in R$ and $y_i$ denotes the parity of the $x_i$. Parity of a number is based on the number of 1 present in it's binary representation. If number of 1 is even then

---

**Algorithm 3** Divisibility tests Data Set Generation Algorithm - $DIV(T, N)$

---

**Require:** $T, N$
  **if** $T$ is "divisible" **then**
    **for** $(\forall i \in [1..N])$ **do**
      $(x_i = [RAND(), RAND()])$
      $y_i = 0$
      $(x_{i,0} \mod x_{i,1} == 0) \rightarrow y_i = 1$
    **end for**
  **end if**
  **if** $T$ is "divisibility_by_3" **then**
    **for** $(\forall i \in [1..N]), x_i = RAND()$ **do**
      $y_i = 0$
      $(x_i \mod 3 == 0) \rightarrow y_i = 1$
    **end for**
  **end if**
  **return** $(x, y)$

---

parity of that will be 0. And if the number of 1 is odd then parity of that will be 1. For example if we take 17. It's binary representation is 10001. It contains 2 one's in its binary representation so parity of 17 is 0. The binary representation data set contains of $D = (x_i, y_i)$ tuples where $x_i \in R$ and $y_i \in 0, 1^K$ where $K$ is 20 in our case denotes its binary representation. For example if $x_i$ is 20 and $K$ is 7 then $y_i$ will be 0010100.

The data set is generated as shown in the schematic (Algorithm 4).

---

**Algorithm 4** Representation Data set Generator- $REP(T, N, K)$

---

**Require:** $T, N, K$
  **if** $T$ is "binary" **then**
    **for** $\forall i \in [1..N]$ **do**
      $x_i = RAND()$ //single number
      $y_i$ is binary representation of $x_i$ in $K$ bits.
    **end for**
  **end if**
  **if** $T$ is "parity" **then**
    **for** $(\forall i \in [1..N])$ **do**
      $x_i = RAND()$
      $s =$ binary representation of $x_i$ in $K$ bits.
      $r =$ count number of 1 in $s$.
      $y_i = 0$
      $(r \mod 2 \neq 0) \rightarrow y_i = 1$
    **end for**
  **end if**
  **return** $(x, y)$

---

## 2.5 DATA SET FOR SORTING PROBLEM

This data set consists of $x_i$ as a list of numbers, mapped to its sorted equivalent $y_i$. The idea is sorting is a very prominent pattern to humans, however basic neurons are not able to capture all of the the element to element relationships. The algorithm for sorting is shown in (Algorithm 5). For example, if $x_i = [1, 3, 2, 1, -9]$ then sorted $y_i = [-9, 1, 1, 2, 3]$.

---

**Algorithm 5** Sorting Test Data Set Generation Algorithm - $SORT(N, K)$

---

**for** $\forall i \in [1..N]$ **do**
   $\forall j \in [1..K] : x_{i,j} = RAND()$
   $y_i = SORT(x_i)$
   /*Sorted version of $x_i$ in ascending order*/
   $(\forall j, k) : (j < k) \rightarrow (y_{i,j} < y_{i,k})$
**end for**
**return** $(x, y)$

---

### 2.6 MACHINE LEARNING MODELS USED IN THE STUDY

#### 2.6.1 DEEP NEURAL NETWORK

The Deep Neural Networks model we used for testing the datasets was of 3 layers each consisting of 100 neurons. This model was implemented in Google Tensorflow(Abadi et al., 2016)

#### 2.6.2 RANDOM FOREST

The Random forest (Liaw et al., 2002) was implemented in Sci-kit Learn(Pedregosa et al., 2011) a Python based library. The configuration in which we use Random forest is (max-depth=5, criterion=$gini$, mini-samples-split=2, n-estimators=10).

#### 2.6.3 RECURRENT NEURAL NETWORK

The RNN model we used for testing the datasets was of three layers. First layer is a LSTM (Gers et al., 1999) cell followed by two layers of RNN cell. We used Relu as activation function. The model was implemented in Google Tensorflow(Abadi et al., 2016).

### 2.7 TRAIN AND TEST DATA SET PARTITIONS

The machine learning model is tested based on a bit lenient train and test partitions of 90% and 10% respectively. The training data size has been increased so as to allow the classifier to make use of more number of patterns as the data sets are tough for any induction based classification formulation today.

## 3 RESULTS

The performance of machine learning models on each of the data sets is computed in two forms - regression and classification. For those problems where the final output is a numeric quantity, the problem is posed as regression and Root Mean Squared Deviation (RMSD) values are reported. The problems posed as regression are selection problems, binary representation of a number, sorting and count of numbers between 0 and 10 and presented in (Table 3). For the problems where the output is a decision, they are posed as classification problem in our setting. The problems posed as classification are divisibility of two numbers, divisibility by 3, parity and detection of duplicate and presented in the (Table 4).

Except for the parity problem where a random forest has performed with $> 92\%$ accuracy, in rest of the places the three methods chosen for evaluation have all failed to give 100% accuracies although the data sets are quite simple.

It is worth discussing, the parity problem in more detail on which random forest appeared to have worked (the * marked observation), however we can readily observe that anything less than 100% would not be of any practical use and not reliable. Consider a binary representation of input of length $K$ elements. The total number of variations of the input is $2^K$. In this case, the input dimensionality is $K = 50$, which corresponds to a data set of $2^{50}$ elements amounting to a total storage cost of $50 \times 4 \times 2^{50} = 200$ *peta bytes*, considering floating point representation for each input element of the vector. When the input dimensionality becomes $1000$, which is not uncommon, the storage requirement for data grows to $1000 \times 4 \times 2^{1000}$ which is an astronomical number. A

| S.No | Dataset Name | RMSD of RF | RMSD of NN | RMSD of RNN |
|------|--------------|------------|------------|-------------|
| 1 | MIN | 7.2788 | 28.2572 | 20.378 |
| 2 | MAX | 7.2788 | 28.398 | 24.397 |
| 3 | TOP2 | 3.912 | 28.455 | 62.81 |
| 4 | BINARY | $\infty$ | $\infty$ | $\infty$ |
| 5 | SORT | $\infty$ | $\infty$ | $\infty$ |
| 6 | COUNT | 10.90 | 28.50 | 4.4312 |

Table 3: Performance of data sets on RNN, NN and RF based on regression formulation. Root Mean Squared Deviation (RMSD) values are computed on the output predictions.

| S.No | Dataset Name | Accuracy of RF | Accuracy of NN | Accuracy of RNN |
|------|--------------|----------------|----------------|-----------------|
| 7 | DIV | 0.691 | 0.5133 | 0.5015 |
| 8 | DIV3 | 0.6668 | 0.6566 | 0.6668 |
| 9 | PARITY | 0.926* | 0.6836 | 0.5279 |
| 10 | DUP | 0.4962 | 0.67 | 0.5427 |

Table 4: Performance of data sets on RNN, NN and RF based on classification formulation. Accuracy values are reported on the output predictions.

decision tree to successfully learn with 100% certainty requires all input patterns to be available in training. The size of the tree in terms of the number of nodes is $2^{K+1} - 1$. Storing a tree having $2^{51} - 1$ nodes has practical limitations for the problem it is trying to address, *as simple as parity* computation. However *a simple for loop* through the data requires only one pass through the input and requires only *a constant space storage* of a couple of floating point variables and predicts with 100% certainty. Having an additional feature in the input, feature engineered to compute parity (i.e. $K + 1$ dimensional input), would reduce the amount of training data requirement to just $K + 1$ points to result in 100% accuracy of any standard classifier.

Each of these problems require at least one pass through the input elements and computing logical inferences. Given multiple possible permutations of the elements of the input vector, any single pass neural network would need to accommodate in some form for those many variations. However, recurrent neural networks do attempt to remember the state

## 4 CONCLUSIONS

We have presented here an issue with limitations of the present day deep neural networks to address deduction based inferences. Today deep neural networks are positioned to do away with feature engineering however we have demonstrated that this is not true. It is an open area of research to configure neural networks to carry out deductive reasoning, however the data sets are mainly sequence of tokens in text where parsing, symbolic unification and deduction abilities come into picture. We have created a simplistic data set that is purely numeric and can be directly consumed by neural networks without a need to learn parsing as well. The notion of knowledge representation as storage of facts and deductive reasoning is not thoroughly captured by the weight space formal representation of neural networks. We have demonstrated failure of typical multi layer perceptron deep neural networks, random forest and recurrent neural network formulations on data sets which require deduction reasoning based features. The data sets are for - (a) selection (3 data sets) - minimum, maximum and top 2nd element in an array of numbers; (b) matching (3 data sets) - duplicate detection, counting and histogram learning; (c) divisibility tests (2 data sets) - divisibility of two numbers and divisibility by 3; (d) representation (2 data sets) - binary representation and parity. We also discussed the parity problem and impracticality of storage and computation to achieve full accuracy by a strong classifier such as a decision tree unless a simple computation friendly feature is engineered. We have presented a case for novel models for *learning deductive reasoning from examples*. These novel models can be a combination of deduction and induction systems operating together as a model on top of an evolutionary computational framework.

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
