# OpenReview forum: "Simple deductive reasoning tests and numerical data sets for exposing limitation of today's deep neural networks"
_ICLR.cc/2021/Conference — Reject_

### Official Review · AnonReviewer2 · 2020-10-13
**Major gaps in related work**

**Rating:** 3
**Confidence:** 5

**Review:**

The paper "Simple deductive reasoning tests and data sets for exposing limitation of today's deep neural networks" describes several datasets for deep learning to test deductive reasoning abilities of neural networks. The paper tests several neural network architectures (as well as random forests) on these datasets and concludes that neural networks are generally not able to perform deductive reasoning.

My main critique is that the authors do not seem to be aware of any of the research that is going on in the area. The opening statement of the paper is as follows: "Learning for Deductive Reasoning is an open problem not yet explicitly called out in the machine learning world today." I'm afraid such a statement is simply not true. Reasoning (including deductive reasoning) is a very active research area in the machine learning communities. See below for a very partial list of works.

Second, the paper contains many assertions that neural networks are incapable of reasoning. For example: "The deep neural networks with today’s notion of a neuron are not suitable for deductive reasoning" The main reason the authors give for this claim is that "neurons" perform only simple arithmetic operations. However, consider that computers only perform simple boolean operations and yet can perform the tasks described in this paper.

These claims are also contradictory to some findings in the literature, which the authors do not seem to be familiar with. Here is a bunch related work that the authors might want to take a look at. (And there are plenty more papers in the area.)

Datasets with similar objectives:

    Nikita Nangia and Samuel R Bowman. Listops: A diagnostic dataset for latent tree learning. arXiv preprint arXiv:1804.06028, 2018.

    "ANALYSING MATHEMATICAL REASONING ABILITIES OF NEURAL MODELS", by Saxton, Grefenstette, Hill, Kohli, ICLR 2019

    "INT: An Inequality Benchmark for Evaluating Generalization in Theorem Proving" Wu, Jian, Ba, Grosse, https://arxiv.org/abs/2007.02924


Datasets and theorem proving with neural networks:

    "DeepMath - Deep Sequence Models for Premise Selection" Alemi et al. https://arxiv.org/pdf/1606.04442.pdf

    "Learning to Prove with Tactics" Gauthier et al. 2018

    "HOList: An Environment for Machine Learning of Higher-Order Theorem Proving", Bansal et al, ICML 2019

    "Learning to Prove Theorems via Interacting with Proof Assistants" Yang, Deng, ICML 2019

    "GamePad: A Learning Environment for Theorem Proving", Huang, Dhariwal, Song, Sutskever, ICLR 2018

    "Generative Language Modeling for Automated Theorem Proving", Polu, Sutskever, arxiv 2020

    "Can Neural Networks Learn Symbolic Rewriting?" Piotrowski et al., 2019

Neural network architectures for reasoning, including (Tree)RNNs, GNNs:

    "Can Neural Networks Understand Logical Entailment?", Evans et al. 2018 https://arxiv.org/abs/1802.08535

    "Graph Representations for Higher-Order Logic and Theorem Proving" Paliwal et al, AAAI 2021

    Also, plenty of pre-deep learning work by Joseph Urban on how to turn logical formulas into features.

Recently, Transformers have been shown to be good at logical reasoning:

    "Deep Learning for Symbolic Mathematics", Lample and Charton, ICML 2020.

    "Transformers Generalize to the Semantics of Logics", Hahn et al, 2020. https://arxiv.org/abs/2003.04218

    "Mathematical Reasoning via Self-supervised Skip-tree Training", Rabe et al, 2020, https://arxiv.org/abs/2006.04757


My third point is that the paper does not specify the experiments precisely. What are the hyperparameters of the neural networks?

Fourth, the paper claims to consider "today's deep neural networks" but does not consider modern neural architectures, such as GNNs and Transformers. These have been shown much better reasoning abilities than RNNs.

In summary: The paper addresses an important question and I encourage the authors to continue to follow this path. But this work does not consider the existing literature at all and a does not make significant contributions beyond the state-of-the-art as far as I can see.


Minor comments:

"A majority of the machine learning models are inductive reasoning models"

I believe by "inductive reasoning" the authors here refer to the learning process. I think the learning phase has to be contrasted with the inference phase.

"However for the sake of convenience and interpretation, a vector is typically represented as a tensor"

The notion of tensor is a generalization of vector.

---

### Official Review · AnonReviewer4 · 2020-10-27
**Great work on developing the deductive reasoning test sets but ignored existing state-of-the-art models and efforts**

**Rating:** 3
**Confidence:** 5

**Review:**

This paper's contribution is introducing a set of tasks and datasets that require deductive approaches as opposed to common induction-based models. The paper tackles an important and interesting problem that helps to shape the future of the neuro-symbolic research area. My main concern however is, the paper ignores and does not cover the current state-of-the-art techniques and their corresponding datasets and by just introducing some datasets fail to give a correct image of the current efforts in this area. For example, the variation of Neural Turing Machine and Memory Networks has been successfully applied to the sorting problem (which has been proposed as one of the tasks of interest in deductive reasoning in this paper as well) [1], however, the authors have not discussed these class of networks at all. In fact, the authors mention the gap in the current models by talking about the need for models that can store the facts and the intermediate results for being able to conduct deductive reasoning but do not talk about the role and shortcomings of Memory Networks and Neural Turing based models or  Neural
Stacks/Queues. Similarly, there are no arguments in the paper about why Neural Theorem Provers [2] cannot be used to emulate the deductive inference mechanism.
In summary, the authors have initiated a good step toward defining the simple deductive reasoning tasks; However, the work has not placed well on the body of current neural and neuro-symbolic techniques, tasks, and datasets and therefore the contribution is not enough for the publication in ICLR.

Minor comments:
- 3rd sentence of the introduction needs rewriting.
- Section 2.2: of of ---> of
- Results: 2^5 0 ---> 2^50


1) Vinyals, Oriol, Samy Bengio, and Manjunath Kudlur. "Order matters: Sequence to sequence for sets." arXiv preprint arXiv:1511.06391 (2015).
2) Rocktäschel, Tim, and Sebastian Riedel. "Learning knowledge base inference with neural theorem provers." Proceedings of the 5th Workshop on Automated Knowledge Base Construction. 2016.

---

### Official Review · AnonReviewer1 · 2020-10-28
**Interesting analysis, but lacking details and unclear motivation**

**Rating:** 4
**Confidence:** 3

**Review:**

*Summary:*
The paper argues that deductive reasoning is an open problem in current machine learning scenarios where features are learned rather than hand-crafted. To highlight the limitations of current approaches, the paper proposes a benchmark suite of 10 simple tasks (finding the minimum, divisibility test, etc.) that are trivial with some feature engineering, but are shown to be very hard without it. Experiments are performed with random forests, neural networks (MLP?), and recurrent neural networks.

*Strengths:*
1. Important to highlight limitations of current neural network based methods. The proposed tasks are very simple for humans, but are discrete and deductive, rather than the inductive setups NNs typically work with.
2. Experiments (although quite limited) show that recent ML approaches exhibit performance close to random.

*Weaknesses:*
1. While I agree that highlighting the drawbacks of current inductive ML approaches is important and that the proposed tasks are hard to do, I don't necessarily see the problem with small feature engineering. Almost every neural approach that is proposed has some engineering - architecture, hyperparameters, data augmentations, etc. that benefit from knowledge about the task or data. For example, CNNs trained on ImageNet use a lot of knowledge: convolutions better than standard linear layers; random crop of the image during train, 5 crops + flips at test time to further improve spatial understanding; image rotation or intensity variation as data augmentation strategies, etc.
2. The paper has a lot of space (is only 6 pages), but does very little to explain the models. No details about the RF, NN, or RNN are mentioned. Is the NN an MLP? How many layers? What are the hidden sizes? How is the RNN used? How is the output produced, last time step hidden state? What is hidden size dimensionality? Details like this matter, and performance metrics without them do not say much.
3. Since the paper takes the stand that feature engineering is key, it would be nice to show improved results with little feature mapping. While it seems that most tasks should be solvable, it is nice to prove that nevertheless. For example, what feature engineering strategy should be used for finding the maximum (when feature engineering already solves the task)? Or would it be enough to represent the real number as a binary sequence for the parity problem?

*Overall rating:*
While the premise is interesting, the work needs to be developed further and presented in much more detail than the current state. In addition, I would like to see some discussion on how some of these deductive reasoning tasks are required as part of an overall intelligent system, rather than just a set of tasks specifically built to break NNs.

*Post-rebuttal*
All reviewers agree that this paper is not up to the mark. While the revision does include several additional related works, they are not very well integrated with the rest of the discussion on the paper. For example, how would some of these memory networks perform? How would Neural Turing Machine do? Considering this, I am hesitant to improve my rating for the paper, even if the collection of related works will certainly help in the re-submission.

---

> ### Author Response · Authors · 2020-11-25
> **We have updated the manuscript keeping in mind what the reviewers suggested.**
>
> Reviewer Comment: I don't necessarily see the problem with small feature engineering. Almost every neural approach that is proposed has some engineering - architecture, hyperparameters, data augmentations, etc. that benefit from knowledge about the task or data.
>
> Response:  We are trying to show that feature engineering is indeed a deductive reasoning mechanism.
> We also demonstrate that machine learning formulation is not suitable to directly operate on trivially simple deduction tasks.
> Although the feature itself may be easy to write code, it brings in huge cost savings in terms of network size and reliability.
>
> Reviewer Comment: The paper has a lot of space (is only 6 pages), but does very little to explain the models.
>
> Response: Our main intention is to demonstrate via simple ‘numeric’ data sets for learning deductive reasoning from examples.
> The existing methodologies on symbolic reasoning data sets are mainly ‘text’ based where it requires the network to learn parsing in addition to inference. They are also complex, requiring to understand theorem and proof steps.
>
> Reviewer Comment: Since the paper takes the stand that feature engineering is key, it would be nice to show improved results with little feature mapping.
>
> Response: In all the problem statements (except sorting), the output itself is the feature. As this would trivially achieve a 100% accuracy or a 0 error,  we have chosen to skip including these numbers.
>
> Reviewer Comment: The work needs to be developed further and presented in much more detail than the current state. In addition, I would like to see some discussion on how some of these deductive reasoning tasks are required as part of an overall intelligent system, rather than just a set of tasks specifically built to break NNs.
>
> Response: We have added a number of citations for the symbolic reasoning efforts in the field to expand the scope of neural networks to encompass deductive reasoning.
> Also, the notion of memory, processor and code how it is attempted to be addressed.
>
> Reviewer Comment: Interesting analysis, but lacking details and unclear motivation
>
> Response: Thanks to the reviewer for valuable inputs.

---

### Official Review · AnonReviewer3 · 2020-10-28
**Simple datasets that are hard to model using neural networks without feature engineering**

**Rating:** 3
**Confidence:** 3

**Review:**

This paper studies the limitations of deep neural networks to model deduction based inferences. This is done by crafting simple datasets and experimentally showing that some (details are not provided) RF, NN and RNN models fail on these.

The paper is hard to follow at places. The main contribution seems to be Algorithms 1-5, which can be used to generate 10 different dataset "benchmarks". The listings of the algorithms seem quite redundant considering the simple types of datasets one wishes to generate, when this would often be achievable using mathematical formula or code "one-liner" (the algorithms are also missing information what is returned and the fonts are used inconsistently). The experimental evaluation gives no details of the trained models.

I agree with the authors, that feature engineering is very relevant when it comes to using ML models and in recent years there has been some tendency to consider neural networks as simple plug-in solutions to all scenarios. However, it seems hardly surprising that the crafted benchmarks proposed here are difficult or even impossible to learn for random forest or neural networks. I might be missing something crucial here, but the paper's contribution seems not really warrant publication.

Pros:
Raising awareness that deep learning is not a plug-in solution for every occasion

Cons:
Significance and novelty seem questionable

Questions:
Please address and clarify the con above

Minor:

This structure is repeated in several places and is hard to parse, consider clarifying it:
" - (a) selection (3 data sets) - minimum, maximum and top 2nd element in an array of numbers; (b) matching (3 data sets) - duplicate detection, counting and histogram learning; (c) divisibility tests (2 data sets) - divisability of two numbers and divisability by 3; (d) representation (2 data sets) - binary representation and parity."

Regarding the statement "However to the best of our knowledge and exploration, today there is no RNN formulation which is meant to learn facts, unification and deductive inferences.", have the authors checked the recent approaches to use deep learning to learn to solve combinatorial problems (e.g., SAT, CSPs) and using GNNs. This is a currently very active area of research that might be interesting to the authors, see e.g.,
https://arxiv.org/abs/1905.13211
https://arxiv.org/abs/1905.12149
https://arxiv.org/abs/1904.01557
https://link.springer.com/chapter/10.1007/978-3-319-98334-9_38
https://openreview.net/forum?id=BJxgz2R9t7
https://openreview.net/forum?id=HJMC_iA5tm
for some recent examples.

---

> ### Author Response · Authors · 2020-11-25
> **We have updated the manuscript keeping in mind what the reviewers suggested.**
>
> Reviewer Comment: The paper is hard to follow at places. The main contributions are the algorithms 1-5.
> “The listings of the algorithms seem quite redundant considering the simple types of datasets one wishes to generate when this would often be achievable using mathematical formula or code "one-liner"
>
> Response: Revised the manuscript to bring in more clarity. The pseudocodes are provided for reproducibility.
>
> Reviewer Comment: The algorithms are also missing information about what is returned and the fonts are used inconsistently.
>
> Response: The issue has been addressed in the revised manuscript.
>
> Reviewer Comment: The experimental evaluation gives no details of the trained models.
>
> Response: The issue has been addressed in the revised manuscript
>
> Reviewer Comment: Significance and novelty seem questionable
>
> Response: Comparison of the proposed data sets against existing symbolic reasoning sets are provided in the introduction section.
>
> Reviewer Comment: Simple datasets that are hard to model using neural networks without feature engineering
>
> Response: Thanks to the reviewer for valuable inputs.

---

### Decision · Program_Chairs · 2021-01-07
**Final Decision**

**Decision:**

Reject

**Comment:**

This paper is not suitable for publication at ICLR. The paper contains a useful message, that neural networks are not a silver bullet, and are especially not well suited to deductive problems. However, as several reviewers pointed out, the claims of the paper are undermined by the fact that it ignores a lot of relevant work on using neural networks in the context of logic reasoning. Reviewer 2 provides a particularly useful list of relevant works on the topic.